# The Mediating Effects of Chronic Diseases in the Relationship Between Adverse Childhood Experiences and Trajectories of Depressive Symptoms in Later Life: A Nationwide Longitudinal Study

**DOI:** 10.3390/healthcare12242539

**Published:** 2024-12-16

**Authors:** Qianqian Dai, Ming Li, Zhaoyu Wang, Qianqian Xu, Xinyi Zhang, Liyuan Tao

**Affiliations:** 1Research Center of Clinical Epidemiology, Peking University Third Hospital, Beijing 100191, China; daisyup0814@163.com (Q.D.); 2311210409@bjmu.edu.cn (Z.W.); 2School of Social Sciences, Tsinghua University, Beijing 100084, China; liming20@mails.tsinghua.edu.cn; 3Institute of Basic Theory for Chinese Medicine, China Academy of Chinese Medical Sciences, Beijing 100700, China; qianqianxu0522@126.com; 4Dongzhimen Hospital, Beijing University of Chinese Medicine, Beijing 100700, China; zxy122570@163.com

**Keywords:** adverse childhood experiences, depressive symptom trajectory, chronic diseases, mediating effects

## Abstract

*Background:* Numerous studies have established a link between adverse childhood experiences (ACEs) and the development of depression in later life. However, the interactive relationships between ACEs, depression, and chronic diseases are still not well understood. In this study, the aim was to investigate the impact of ACEs on depressive trajectories among middle-aged and elderly individuals in China, as well as to examine the mediating roles of chronic diseases in this association. *Methods:* Data were drawn from 6921 participants aged 45 and older, using the China Health and Retirement Longitudinal Study (CHARLS) data from 2011, 2013, 2015, and 2018, combined with the 2014 life history survey. Depressive symptom scores were assessed using the widely recognized CES-D-10 scale. The trajectories of depressive symptoms were identified via group-based trajectory modeling (GBTM). The association between ACEs and depressive trajectories was analyzed using multinomial logistic regression, and the KHB method was employed to test the mediating effects of different chronic diseases. *Results:* The age of the 6921 participants was 57.2 ± 8.0 years, with females comprising 53.9% and males 46.1%. We found that approximately 70% of Chinese middle-aged and older adults had experienced at least one ACE, and 4.8% had experienced four or more ACEs. The following four distinct trajectories of depressive symptoms were identified: continuing-low (N = 1897, 27.4%), continuing-low-to-middle (N = 2937, 42.4%), continuing-middle-to-high (N = 1649, 23.8%), and continuing-high (N = 438, 6.3%). Compared to individuals without ACEs, those with four or more ACEs had a significantly higher likelihood of following the continuing-low-to-middle trajectory (OR = 2.407, 95%CI: 1.633–3.550), the continuing-middle-to-high trajectory (OR = 7.458, 95%CI: 4.999–11.127), and the continuing-high trajectory (OR = 20.219, 95%CI: 12.115–33.744), rather than the continuing-low trajectory. Exposure to a greater number of ACEs was associated with an increased risk of following an adverse trajectory of depressive symptoms. Multiple chronic diseases significantly mediated the relationship between ACEs and depressive trajectories, with arthritis or rheumatism exerting the largest mediating effect, followed by digestive and respiratory diseases. *Conclusions:* These findings indicated that ACEs were associated with a higher risk of worse depressive symptom trajectories, with different chronic diseases mediating this relationship. Therefore, developing public measures to prevent ACEs can reduce the risk of chronic diseases and depression in middle-aged and elderly people. Additionally, strengthening the prevention and management of chronic diseases in individuals exposed to ACEs may further reduce their subsequent risk of depression.

## 1. Introduction

Depression is one of the most prevalent mental disorders, and it has become the leading cause of disability worldwide [1]. The lifetime prevalence of depression in Chinese adults is approximately 6.8%, leading to significant social and functional impairment [2], while the overall prevalence of depressive symptoms among elderly people in China is 23.6%, and it is still exhibiting a sustained upward trend [3]. Depression severely limits psychosocial functioning, diminishes the quality of life, and imposes a significant social burden. As a major public health issue, it has led to an escalating burden of disease, particularly within the elderly population [4]. However, the treatment rate for depression is alarmingly low, with many individuals failing to receive adequate therapy [2]. A study reported that in low- and middle-income countries, over 75% of individuals with depression are unable to obtain treatment [5]. Consequently, there is an urgent need to identify risk factors and potential mechanisms associated with depression to facilitate the timely implementation of preventative and interventional strategies. The etiology of depression is multifactorial, involving a complex interplay of biological, psychological, and social factors, with the origins of its risk factors potentially traceable to early life experiences from a life course perspective [6,7].

Adverse childhood experiences (ACEs) encompass a wide array of stressful, sudden, or traumatic events that occur during childhood, including abuse, neglect, and household dysfunction, and so on, also known as childhood adversity [8]. ACEs are very common in both developed and developing countries and can exert long-term detrimental effects on health in middle and later life from the life-course perspective [9]. A cross-sectional survey involving 9508 adults revealed that over 50% of respondents reported at least one ACE while 25% reported two or more ACEs. Individuals who had four or more ACEs exhibited a 4- to 12-fold increased risk for alcohol abuse, drug use, depression, and suicide compared to those without such experiences, and the number of ACEs also showed a graded relationship with heart disease, cancer, chronic lung disease, liver disease, and so on [8]. Additionally, a large body of evidence has demonstrated that individuals with negative childhood experiences are at a higher risk of depression in middle and old age, and there is a dose–response relationship between cumulative ACEs and subsequent depression [10,11,12,13,14]. However, few studies have used trajectory models to explore the dynamic trends of depressive symptoms over time and the impact of ACEs on depressive trajectories, and the underlying mechanisms of this connection, remain elusive.

Childhood is a critical period for brain development and children who have experienced various forms of ACEs are at an increased risk of difficulty in brain development, further affecting cognitive and emotional regulation capabilities [15]. Many clinical studies have shown that depression is associated with a highly dysregulated hypothalamic–pituitary–adrenal (HPA) axis and abnormal amygdala function [16,17,18]. Numerous animal experiments have also demonstrated that early experiences and prenatal stress have a certain impact on the enlargement of the amygdala and the shrinkage of the hippocampus [19,20]. The stimulation of neurogenesis in the hippocampus is crucial for alleviating depression [21]. In fact, beyond mental health, adverse social and contextual experiences during childhood have a lasting impact on the physical health of adults. Stressful childhood experiences result in persistent proinflammatory phenotypes in immune cells, which may lead to low-grade systemic inflammation and subsequently cause the development and progression of multiple chronic diseases [22]. The stress response may suppress immune function, and in some cases, enhance it. It may also communicate bidirectionally with the central nervous and endocrine systems through the immune system, causing dysbiosis and inflammation of the gut microbiota, thereby affecting human health [23,24,25]. Exposure to ACEs has been found to be related to the risk of various types of chronic diseases, including but not limited to chronic lung disease, coronary heart disease, stroke, diabetes, digestive system disease, arthritis, as well as multimorbidity [26,27,28,29,30]. In addition, the association between chronic diseases and depressive symptoms has also been substantiated. A substantial quantity of studies in the literature have found that depression is a common and burdensome complication of chronic diseases [31,32,33]. Understanding whether chronic diseases serve as mediators in the relationship between ACEs and depression over time, and to what extent they influence this association, would contribute to enhancing the preventative and therapeutic measures for depression and to pointing out the direction for the formulation of public health strategies.

Previous studies have indicated that ACEs can increase the risk of depression in middle-aged and elderly individuals [10,11,12,13,14], and many studies have shown an association between ACEs and various chronic diseases [26,27,28,29,30]. Currently, there is no research exploring whether different types of chronic diseases mediate the relationship between ACEs and depression trajectories. A study found that multimorbidity plays a mediating role between ACEs and depression in middle-aged and elderly people, but it did not clarify the relationship between ACEs and depression trajectory, and it did not consider the mediating effects of different types of chronic diseases [34]. Therefore, building on the existing research base, this study was designed to explore the association between ACEs and longitudinal patterns of depressive symptom trajectories, while also examining the mediating roles of chronic diseases in this relationship by using a nationally representative cohort of Chinese middle-aged and elder adults. Based on the presented scientific literature, we hypothesized that exposure to a higher number of ACEs would be associated with an increased risk of worsening depressive symptom trajectories, and that different chronic diseases would serve as mediators in the relationship between ACEs and the progression of depression over time.

## 2. Methods

### 2.1. Participants

The data for this study were drawn from the China Health and Retirement Longitudinal Study (CHARLS), a nationally representative longitudinal survey of middle-aged and elderly community-dwelling adults [35,36]. In brief, the national baseline survey of CHARLS was conducted between 2011 and 2012, covering 17,708 respondents randomly selected across 150 counties or districts and 450 villages or urban communities in 28 provinces in China. To date, several follow-up surveys were performed in 2013 (N = 18,612), 2015 (N = 21,097), 2018 (N = 19,816), and 2020 (N = 19,395). In 2014, face-to-face interviews were conducted to collect life history information for all living respondents in the 2011 and 2013 waves (N = 20,656). In this study, we utilized data from the 2011, 2013, 2015, and 2018 main surveys and the 2014 life history survey. The final analytical sample included 6921 individuals aged 45–87 years after excluding those without the required information on any analytical measures (Figure 1).

### 2.2. Measurements of Depressive Symptoms

Depressive symptoms were assessed repeatedly using ten-item Center for Epidemiological Studies Depression Scale (CES-D-10) in each wave [37,38]. As a validated tool with adequate reliability and validity, CES-D-10 has been widely used to measure the degree of depressive symptoms in middle-aged and elderly people in China [39]. The score for each item ranged from 0 to 3, and all these ten items were summed to create a cumulative score, ranging from 0 to 30. The higher the scores, the higher the level of depressive symptoms. The specific items for calculating CES-D-10 are shown in Appendix A.

### 2.3. Definition of ACEs

In the 2014 life history survey, relevant information on ACEs before the age of 17 was retrospectively collected in the self-reported life history module. According to previous studies [40,41,42], we extracted 10 widely used ACE indicators, including five threat-related ACEs (physical abuse, household substance abuse, domestic violence, unsafe neighborhood, and bullying) and five deprivation-related ACEs (emotional neglect, household mental illness, incarcerated household member, parental death, and parental separation or divorce). Detailed questionnaires and measurements of ACE indicators are available in Appendix A. Each item of ACEs was dichotomized (0 for absent or 1 for present) and summed up to generate a cumulative ACE score, ranging from 0 to 10. with higher scores indicating more severe childhood adversity. Participants were further grouped into the following five categories according to the cumulative ACE score: 0, 1, 2, 3, and ≥4 ACEs.

### 2.4. Assessment of Chronic Diseases

Chronic disease information was collected from the 2011 baseline survey of CHARLS. We ascertained whether respondents had chronic diseases through self-reports of physician-diagnosed conditions from the survey. Based on previous research, we evaluated 10 common noncommunicable chronic diseases and grouped them into the following four categories based on the affected organs and systems: digestive diseases (stomach or liver disease), respiratory diseases (chronic lung disease or asthma), arthritis or rheumatism, and cardio-metabolic diseases (hypertension, dyslipidemia, diabetes, heart disease, or stroke) [43]. Respondents who reported any chronic diseases within each category were identified as having the condition. Each indicator was constructed into a dichotomous variable, with 1 for present or 0 for absent.

### 2.5. Covariates

We controlled for multiple confounding factors to minimize bias in the association between ACE and depressive symptoms on the basis of previous studies. The data regarding the following covariates also came from the 2011 baseline survey of CHARLS. Demographic characteristics included gender (male or female), age, marital status (married/cohabiting or unmarried/separated), Chinese household registration hukou status (agricultural hukou or no-agricultural hukou), and current residence (urban or rural). Socioeconomic status characteristics included participants’ education level (no formal education, elementary school, middle school, or high school and above), employment status (agricultural employed, non-agricultural employed, retired, or unemployed). We also considered parental education level, measured by parents’ highest level of education (illiteracy, or primary school and above) to measure participants’ childhood socioeconomic status. Drinking status (never drink, abstainer, or current drinker) and smoking status (never smoke, former smoker, or current smoker) as two indicators of health-related behaviors were included in our study. Except for setting age as a continuous variable, all other covariates were set as categorical variables.

### 2.6. Statistical Analysis

Firstly, we conducted descriptive statistics to analyze the participants’ basic characteristics, ACE scores, prevalence of each ACE item, and distribution of chronic diseases.

Next, the group-based trajectory model (GBTM) was employed to identify participants with similar longitudinal patterns trajectories of depressive symptoms during the 7-year follow-up period [44]. We established five trajectory models featuring 5, 4, 3, 2, and 1 trajectory, and then compared fit indices of these models based on the model selection criteria of GBTM. The optimal trajectories were chosen based on the Akaike information criterion (AIC), the Bayesian information criterion (BIC), and the average posterior probability (AvePP). AIC and BIC are commonly used to compare the goodness of fit of models. AvePP is an indicator of the mean of the probabilities that each individual belongs to the groups they are assigned to by the model, where a value above 0.7 should be achieved [44,45].

In the third step, multinomial logistic regression was further employed to explore the relationships between ACEs and depressive symptom trajectories after adjusting for potential confounding factors. We set the continuing-low trajectory as the reference group. Model 1 was adjusted for gender, age, education level, marital status, hukou status, residence, parental education level, participants’ education level, participants’ employment status, smoking, and drinking in the 2011 baseline survey, while model 2 additionally included the mediators. Odds ratios (OR) and 95% confidence interval (CI) were reported for regression models.

Then, the Karlson–Holm–Breen (KHB) mediation analysis method was applied to examine the mediating effects between ACEs and depressive trajectories [46,47]. We estimated the total impact of ACEs on the trajectories of depressive symptoms, as well as its direct effect and indirect effect and the contribution of each type of chronic diseases to the overall mediating effects was calculated.

Finally, we conducted a series of sensitivity analyses to test the robustness of our findings. In brief, we re-examined the relationship between ACE scores and depressive symptom trajectories, as well as the mediating effects of different chronic diseases, by excluding participants with memory-related diseases at the 2011 baseline and treating ACE scores as continuous variables.

Categorical and continuous variables were presented as frequency (percentage) and mean (standard deviation, SD), respectively. The Chi-Square (χ^2^) test or the analysis of variance F-test were employed to assess the differences across various ACE groups and depressive symptom trajectories. All statistical analysis were performed using Stata version 18.0. A two-tailed *p* value < 0.05 was considered statistically significant.

## 3. Results

### 3.1. Descriptive Analysis

Baseline characteristics of populations with different ACE scores are presented in Appendix A. A total of 6921 middle-aged and elderly adults were recruited in this study, with an average age of 57.2 ± 8.0 years, comprising 3188 (46.1%) males and 3733 (53.9%) females. Most of the participants were married or cohabiting (90.8%), had an agricultural hukou (82.4%), and resided in rural areas (65.6%). As is shown in Figure 2, approximately 70% of participants reported at least one ACE, and 4.8% of participants reported exposure to at least four ACEs. The prevalence ranged from the lowest at 0.3% (for incarcerated households) to the highest at 32.8% (for emotional neglect) among the 10 ACE items. There were significant differences among individuals with different ACE scores in terms of gender, age, parental education level, participants’ education level, drinking status, smoking status, digestive diseases, respiratory diseases, and arthritis or rheumatism. But there was no difference in marital status, hukou status, current residence, employment status, and cardio-metabolic diseases. Compared with individuals without ACEs, those who reported ACEs were more likely to have a low educational level, either for themselves or for their parents. A higher ACE score correlated with an increased prevalence of digestive diseases, respiratory diseases, and arthritis or rheumatism. These findings were consistent with previous research findings [40,41].

### 3.2. Identification of Depressive Symptoms Trajectories

Based on GBTM, we tested how many trajectories of depressive symptoms were optimal to explain the heterogeneity in the depressive symptoms scores over a 7-year span. The fit statistic indices are reported in Appendix A. According to the selection criteria of the GBTM model, we identified four distinct depressive symptom trajectory memberships. The AvePP values of four trajectory groups were 0.78, 0.73, 0.80, and 0.88 with each group representing more than 5% of the total sample. As presented in Appendix A, the four longitudinal trajectories of depressive symptom scores plotted by survey years were as follows: continuing-low (N = 1897, 27.4%), continuing-low-to-middle (N = 2937, 42.4%), continuing-middle-to-high (N = 1649, 23.8%), and continuing-high (N = 438, 6.3%).

The baseline characteristics of respondents in each class of depressive symptom trajectories are reported in Table 1. There was a significant relationship between ACE scores and depressive symptom trajectories. Compared to participants who were in the continuing-low trajectory (average CES-D-10 score consistently around 3), individuals in the continuing-low-to-middle trajectory (average CES-D-10 score consistently around 7), continuing-high trajectory (average CES-D-10 score consistently around 13), and high-stable trajectory (average CES-D-10 score consistently around 20) were more likely to be older, female, of unmarried or separated status, have agricultural hukou, live in rural areas, work in agriculture, have lower levels of education for both themselves and their parents, as well as having experienced multiple ACEs and suffer from multisystem chronic diseases. Furthermore, the results showed that there were significant differences between the participants in each trajectory in terms of all covariates and four categories of chronic diseases. Subsequently, covariates that demonstrated statistical significance were incorporated into the multinomial logistic regression model for further analysis.

### 3.3. Associations of ACEs and Depressive Symptom Trajectories

We considered the continuing-low trajectory as a reference and estimated multinomial logistic models to investigate relationships between ACEs and depressive symptom trajectories. All models were adjusted for potential confounders, including gender, age, marital status, hukou status, current residence, education level, employment status, parental education level, drinking status, and smoking status (Model 1). As shown in Table 2, in comparison to middle-aged and elderly adults who did not experience childhood adversity, individuals who experienced one or more ACEs had a significantly higher likelihood of being in continuing-low-to-middle trajectory (for 1 ACE, OR = 1.252, 95%CI: 1.084–1.446; for 2 ACEs, OR = 1.450, 95%CI: 1.221–1.723; for 3 ACEs, OR = 1.948, 95%CI: 1.524–2.488; for ≥4 ACEs, OR = 2.407, 95%CI: 1.633–3.550), continuing-middle-to-high trajectory (for 1 ACE, OR = 1.482, 95%CI: 1.240–1.771; for 2 ACEs, OR = 2.081, 95%CI: 1.694–2.557; for 3 ACEs, OR = 3.899, 95%CI: 2.966–5.126; for ≥4 ACEs, OR = 7.458, 95%CI: 4.999–11.127), and continuing-high trajectory (for 1 ACE, OR = 2.015, 95%CI: 1.450–2.800; for 2 ACEs, OR = 4.816, 95%CI: 3.439–6.745; for 3 ACEs, OR = 7.836, 95%CI: 5.191–11.827; for ≥4 ACEs, OR = 20.219, 95%CI: 12.115–33.744), rather than continuing-low trajectory. Exposure to more ACEs was associated with an increased risk of entering a higher score in the depressive symptom trajectory. These results indicated adverse events in early life dramatically increased the likelihood of developing depressive disorders in middle and late adulthood.

In Model 2, we further introduced chronic diseases as potential mediators between ACEs and depressive symptom trajectories. The results indicated that participants with any chronic diseases have an increased likelihood of entering a worse trajectory of depressive symptoms compared to those without chronic diseases. In addition, the association between ACEs and depressive symptom trajectories has been attenuated by different chronic diseases but remained significant. This preliminarily finding suggested that chronic diseases may play partial mediating effects in the pathways underlying the relationship between ACEs and depressive symptom trajectories. Figure 3 showed the association between ACE scores and depressive symptom trajectories, both before and after introducing chronic diseases.

### 3.4. The Mediating Effects of Chronic Diseases

The analysis results of KHB mediation method were reported in Table 3. The total effect of exposure to any ACEs on depressive symptom trajectories were decomposed into direct effect and indirect effect through four classes of chronic diseases. We further calculated the proportion of the total indirect effect explained by each mediator. The result showed that the indirect influence of ACEs on depressive symptom trajectories significantly mediated by different chronic diseases was estimated at the range of 12.72–20.93%. This indirect influence was mainly contributed by digestive diseases, arthritis or rheumatism, and respiratory diseases, while cardio-metabolic diseases could not mediate the relationship. Specifically, the OR for indirect effect of exposure to four or more ACEs on depressive symptom trajectories (continuing-low-to-middle vs. continuing low) through chronic diseases was 1.224 (95%CI: 1.126–1.330), accounting for 20.93% of the total indirect effect, with digestive diseases, arthritis or rheumatism, and respiratory diseases accounting for 6.33%, 9.89%, and 4.52% of the total effect, respectively. The OR for the indirect effect of exposure to four or more ACEs on depressive symptom trajectories (continuing-middle-to-high vs. continuing low) through chronic diseases was 1.464 (95%CI: 1.270–1.688), accounting for 17.59% of the total indirect effect, with digestive diseases accounting for 6.16%, arthritis or rheumatism accounting for 8.02%, and respiratory diseases accounting for 3.28% of the total effect, respectively. The OR for the indirect effect of exposure to four or more ACEs on depressive symptom trajectories (continuing-high vs. continuing low) through chronic diseases was 1.792 (95%CI: 1.454, 2.208), accounting for 18.03% of the total indirect effect, with digestive diseases accounting for 6.19%, arthritis or rheumatism accounting for 7.56%, and respiratory diseases accounting for 4.19% of the total effect, respectively.

### 3.5. Sensitivity Analysis

In the sensitivity analysis, we performed a series of examinations to verify the reliability of our results. On the one hand, we re-examined the association between ACEs and the trajectories of depressive symptoms by excluding participants who had memory-related diseases at the 2011 baseline. This step was taken to reduce recall bias, and the results remained robust, as detailed in Appendix A. On the other hand, we repeated the above data analyses using the ACE score as a continuous variable, which yielded results that were consistent with those obtained when the ACE score was regarded as a categorical variable, as presented in Appendix A. These sensitivity analyses collectively strengthen the reliability of our findings.

## 4. Discussion

In this large nationally representative cohort analysis, we investigated the relationship between ACEs and depressive symptom trajectories in mid- and later-life among Chinese adults and analyzed the potential mediating effects of different chronic diseases. We found that among the 6921 Chinese adults aged 45 and over from CHARLS 2011 to 2018, approximately 70% of respondents had experienced at least one ACE, and 4.8% of respondents had experienced at least four ACEs. Compared to individuals without an ACE, those exposed to ACEs were more likely to enter the continuing-low-to-middle trajectory, continuing-middle-to-high trajectory, and continuing-high trajectory, rather than the continuing-low depressive symptom trajectory. Exposure to a higher number of ACEs was positively associated with more unfavorable depressive symptom trajectories in middle-aged and elderly individuals. Moreover, people with chronic diseases were more likely to develop more severe depressive symptoms. Chronic diseases emerged as substantial mediators in the relationships between ACEs and depressive symptom trajectories, with the most pronounced effects attributed to arthritis or rheumatism, succeeded by digestive disorders, and respiratory conditions. Our study emphasized the persistent impact of ACEs on depression in later life, with different chronic diseases playing significant mediating roles in this relationship. These results revealed two crucial opportunity windows for mitigating depression, with one for taking preventative measures to lower the incidence of ACEs, and another for implementing interventions targeting different chronic diseases in later life.

### 4.1. Mechanisms of ACEs Leading to Depression and Chronic Diseases

Our findings were consistent with the prior research that ACEs increased the risk of subsequent depressive symptoms [10,11,12,13,14]. A recent study involving 25,252 twins found that the correlation between ACEs and adult mental health outcomes still existed after accounting for shared genetic and environmental influences, particularly after multiple ACEs or sexual abuse [48]. Several plausible explanations have been posited regarding the association between ACEs and subsequent depressive outcomes. The stressors throughout the entire life course were closely associated with the occurrence and exacerbation of depression [49]. Early traumatic experience alters the response of neurogenesis to stress later in life [50,51]. Compared to individuals without childhood abuse, the areas of the intracranial, cerebral, prefrontal cortex, prefrontal cortical white matter, and right temporal lobe of abused children were smaller, which may be a potential pathogenesis of depression induced by ACEs [52,53].

Furthermore, the psychological stress during childhood and adolescence is closely related to the inflammatory response throughout the entire life course [54]. Neurobiological studies showed that ACEs lead to a weakened cortisol response to psychosocial stress, an elevated inflammatory response, and an enhanced amygdala response to negative emotional stimuli, thereby affecting physical and mental health [55]. The theory of biological embedding posited that ACEs induce substantial biological changes in children, damaging the HPA axis, leading to chronic inflammation, and disrupting hemodynamic and autonomic functions [56]. From the perspective of sociological cumulative inequality theory, the disadvantages stemming from early life adversities tended to accumulate across the life span. Such initial setbacks can impede individuals’ access to essential social resources and opportunities, including education, career prospects, and socioeconomic status, thereby potentially elevating the risk of health issues later in life [57,58,59]. Toxic stress disrupted the fundamental elements of optimal health and development in early childhood, and it is crucial to protect the developing brain and other biological systems from the physiological damage caused by toxic stress [60].

### 4.2. Chronic Diseases Play Mediating Effects Between ACEs and Depression Trajectories

A series of studies has confirmed that the relationship between ACEs and depressive symptoms is mediated by certain factors, such as cognition, activities of daily living (ADL), inflammation, sleep duration, marital status, etc. [61,62,63,64]. In addition to these factors, complex health conditions may also serve as important mediators in the association between ACEs and depressive trajectories. Individuals who have experienced multiple adversities in childhood were also more likely to encounter a variety of health risks later in life. A nationwide cross-sectional study evaluated the association between 12 types of ACEs and 14 non-communicable chronic diseases, revealing a dose–response relationship between the quantity of ACEs and the risk of the majority of chronic diseases [29]. Consistent with previous research, in our study, we found that the higher the ACE score, the higher the incidence of chronic diseases, and that different chronic diseases played a mediating role in the relationship between ACEs and depression. Our results were also supported by a prior investigation that the risks of multimorbidity and depression associated with ACEs continually increased across the lifespan, and multimorbidity serving as a mediator in the relationship between ACEs and depression among middle age and older adults [34]. This research used parallel process latent growth curve modelling to analyze the association of ACEs, multimorbidity, and depression, focusing solely on the presence of two or more chronic noncommunicable diseases simultaneously, without taking into account the specific types of diseases or elucidating the mediating roles of different types of chronic diseases.

### 4.3. Mediating Roles of Different Types of Chronic Diseases

Our mediation analysis further indicated that arthritis or rheumatism, digestive diseases, and respiratory diseases significantly mediated the relationship between ACE scores and depressive symptom trajectories, with arthritis or rheumatism having the greatest mediating effects. This might be attributable to the fact that exposure to ACEs can have long-term effects on individuals’ immune system function, increasing the risk of arthritis or rheumatism later in life, consequently leading to depression in middle and old age [65,66]. Notably, our research indicated that disorders of the digestive system explained 30% to 47% of the indirect influences on the progression of depressive symptoms related to different ACE scores. This was in line with previous findings that have established a correlation between ACEs and the heightened likelihood of digestive system illnesses [67]. A number of studies have demonstrated that ACEs are associated with the development of digestive system abnormalities such as irritable bowel syndrome, gastrointestinal symptoms, and gut–brain interaction disorders [68,69,70]. These gastrointestinal diseases can further lead to depression through the HPA axis, intestinal permeability, gut microbiota, and inflammation [71]. Additionally, individuals with gastrointestinal dysfunction may experience abnormalities in brain regions, such as significant activation of the amygdala, which increases the risk of depression [72]. Our research additionally determined that respiratory diseases contributed approximately 1% to 23% to the indirect influences on the relationship between varying ACE scores and depressive symptom trajectories. Similarly, multiple studies have validated the link between exposure to ACEs and an increased likelihood of chronic lung disease development [73,74], which is also associated with the risk of depression progression [75,76]. However, we found that cardio-metabolic diseases did not play mediating effects in the relationship between ACEs and depressive trajectories. Consistent with our outcomes, a prior investigation also showed that ACE scores did not have a substantial link to the risk of hypertension and diabetes [29]. The intrinsic mechanisms warrant additional investigation and clarification. Moreover, even after incorporating chronic diseases as mediators, substantial correlations persisted between ACEs and depressive symptom trajectories, underscoring the necessity for additional exploration into the underlying mechanisms linking these factors in the future.

### 4.4. Strengths and Limitations

A strength of this study was that it was conducted on a large, national sample with a long follow-up period, aiming to explore the associations between ACEs, different chronic diseases, and depressive trajectories in middle-aged and older Chinese adults, thereby ensuring the representativeness and credibility of the results. Secondly, considering the dynamic and intermittent characteristics of depressive symptoms, it is insufficient to depend solely on a single measure or a cross-sectional analysis to uncover the genuine correlation between ACEs, chronic diseases, and depression. We utilized the GBTM to track the progression of depressive trajectories, providing more accurate information on the changes in depression symptoms over time than a single time point assessment, and thus may be more valuable. Thirdly, from the life course perspective, our mediation analysis provided new insights for the impact of ACEs on various types of chronic diseases and the subsequent development of depression and suggested that preserving physical well-being could be a key intervention approach for mitigating mood disorders. Public health initiatives focused on managing multiple chronic diseases, especially arthritis or rheumatism, digestive diseases, and respiratory diseases among middle-aged and elderly individuals, are crucial strategies for addressing depression among those who have exposed ACEs.

Several limitations also need to be considered in relation to our study. Firstly, due to the retrospective design of this study, there may be some recall bias in the ACE entries reported by participants. However, a study investigated the reliability of retrospective ACE assessments and determined that recall bias is unlikely to significantly reduce the accuracy of outcome measurements, so we assumed that our retrospective data remained valid [77]. Further investigation is warranted to ascertain whether prospective and retrospective ACEs exhibit distinct correlations with chronic diseases and depressive symptoms [78]. Additionally, after conducting sensitivity analysis through excluding participants who had memory-related diseases at the 2011 baseline, we found that ACEs are still significantly associated with chronic diseases and depression, demonstrating the robustness of our results. Secondly, most participants were excluded from our research due to missing data, which may result in selection bias to some extent. Excluding missing values is inevitable in longitudinal studies, and considering the completeness of the research data, this may also be a reasonable approach. Thirdly, the measurement of chronic diseases was based on data from the 2011 baseline survey, and our study did not take into account the population diagnosed with chronic diseases from 2011 to 2018. In fact, new-onset chronic diseases may lead to a deterioration in mental health, thereby affecting the trajectories of depressive symptoms [79]. Further in-depth research is needed to clarify these causal relationships in the future. Finally, the severity of trauma caused by different types of ACEs may vary. However, we did not take into account the weight of each ACE entry, instead assuming that their risks were equal and accumulated ACE scores were used for statistical analysis, which may affect the accuracy of the measurements [80]. Meanwhile, we did not incorporate an assessment of sexual abuse, which is one of the most traumatic ACEs that children may endure. The assessment of this factor was precluded owning to the absence of requisite data. We repeated the above data analysis using the ACE score as a continuous variable, and the results remained consistent with the previous findings, further suggesting the reliability of our research.

### 4.5. The Impactions of ACEs on Physical and Mental Health

In this study, the relationship is disclosed between ACEs and the depression trajectory among middle-aged and elderly individuals in China, and probed into the mediating role of various chronic diseases. This discovery holds profound implications for formulating public health policies and enhancing clinical practice. Although the causes of depression among middle-aged and elderly people can be partly ascribed to the onset of chronic diseases, their risk factors can frequently be traced back to childhood. Hence, in this study, the necessity was emphasized for all sectors of society to pay attention to the environment in which children grow up and the government was called upon to formulate relevant policies to shield children from the harm of ACEs, which are of vital importance for their long-term physical and mental well-being. Given the aging population and the escalating incidence of chronic diseases, it is anticipated that the incidence rate of depression among the middle-aged and elderly will continue to rise. Our research provided a scientific basis for identifying potential prevention and intervention targets for depression. Strengthening the management of chronic diseases, such as arthritis, rheumatoid arthritis, digestive system diseases, and respiratory system diseases among middle-aged and elderly people, and developing comprehensive medical service measures is of great significance for alleviating the psychological trauma caused by ACEs and reducing the risk of depression in later life.

## 5. Conclusions

In this study, a significant positive link was identified between ACEs and depressive symptom trajectories in middle and later life among Chinese adults, with chronic diseases mediating this relationship. Notably, arthritis or rheumatism had the largest mediating effect, followed by digestive and respiratory diseases. Public health policies and behavioral interventions targeting the prevention and mitigation of ACEs could provide children with a safe and stable environment for growth, which potentially reduces the burden of physical diseases and mental disorders in adulthood. From a comprehensive life-course perspective, we can better understand and prevent the physical and mental health problems that stem from ACEs. Public health measures aimed at reducing chronic diseases such as arthritis, rheumatism, gastrointestinal diseases, and pulmonary diseases are an important way of alleviating depression in Chinese middle-aged and elderly people affected by ACEs. In addition to the primary prevention of ACEs in early life, strengthening the detection and management of various chronic diseases in later life is equally important for improving depression symptoms in middle-aged and elderly people.

## Figures and Tables

**Figure 1 healthcare-12-02539-f001:**
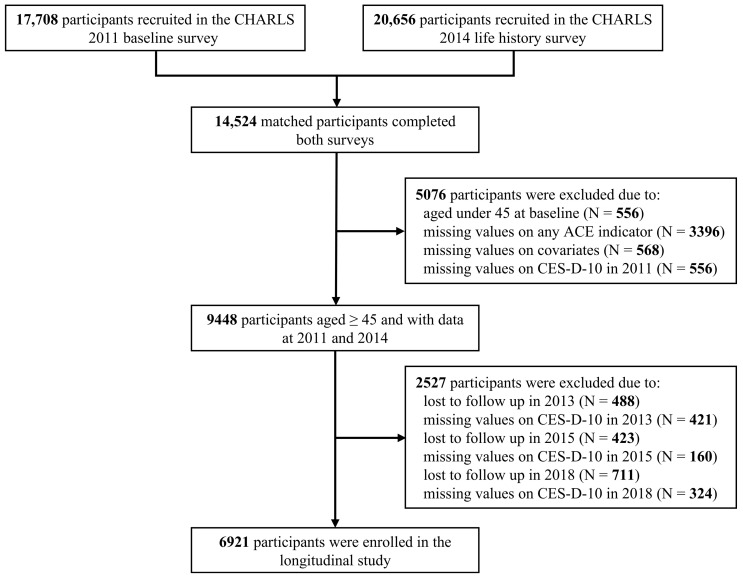
Flowchart of the participant selection.

**Figure 2 healthcare-12-02539-f002:**
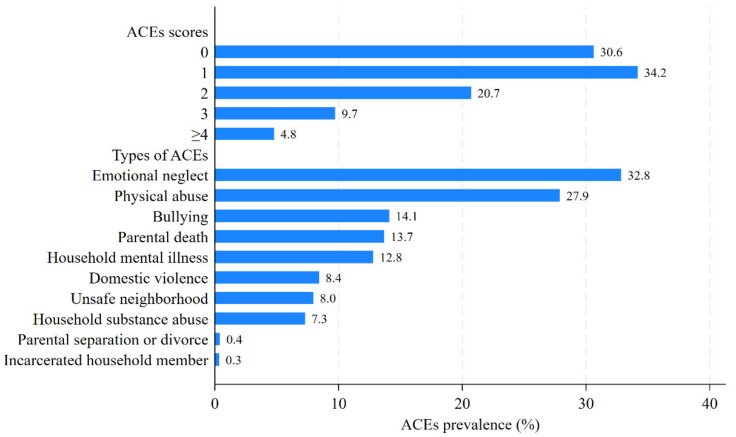
The percentage of participants with different ACE scores and ACE types.

**Figure 3 healthcare-12-02539-f003:**
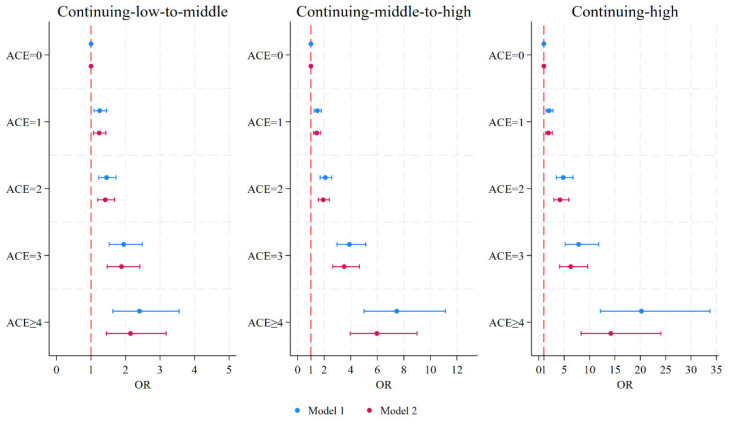
Forrest plot of the association between ACE scores and depressive symptom trajectories. Model 1 was adjusted for gender, age, education level, marital status, hukou status, residence, parental education level, participants’ education level, participants’ employment status, smoking, and drinking in 2011 baseline survey. Model 2 additionally included the mediators. ACEs, adverse childhood experiences; OR, odds ratio.

**Table 1 healthcare-12-02539-t001:** Baseline characteristics of participants with distinct depressive symptom trajectories.

Variables	Total Sample (N = 6921)	Depressive Symptom Trajectories				*p*-Value
		Continuing-Low (N = 1897)	Continuing-Low-to-Middle (N = 2937)	Continuing-Middle-to-High (N = 1649)	Continuing-High (N = 438)	
ACE scores, N (%)						<0.001
0	2119 (30.6)	711 (37.5)	929 (31.6)	412 (25.0)	67 (15.3)	
1	2365 (34.2)	674 (35.5)	1044 (35.5)	530 (32.1)	117 (26.7)	
2	1433 (20.7)	349 (18.4)	594 (20.2)	359 (21.8)	131 (29.9)	
3	673 (9.7)	123 (6.5)	265 (9.0)	217 (13.2)	68 (15.5)	
≥4	331 (4.8)	40 (2.1)	105 (3.6)	131 (7.9)	55 (12.6)	
Age (years), (mean ± SD)	57.2 ± 8.0	56.2 ± 7.9	57.6 ± 8.2	57.4 ± 7.9	58.6 ± 7.5	<0.001
Gender, N (%)						<0.001
Male	3188 (46.1)	1274 (67.2)	1307 (44.5)	512 (31.0)	95 (21.7)	
Female	3733 (53.9)	623 (32.8)	1630 (55.5)	1137 (69.0)	343 (78.3)	
Marital status, N (%)						<0.001
Married/cohabiting	6284 (90.8)	1797 (94.7)	2668 (90.8)	1461 (88.6)	358 (81.7)	
Unmarried/separated	637 (9.2)	100 (5.3)	269 (9.2)	188 (11.4)	80 (18.3)	
Hukou status, N (%)						<0.001
Agricultural hukou	5702 (82.4)	1423 (75.0)	2415 (82.2)	1467 (89.0)	397 (90.6)	
Non-agricultural hukou	1219 (17.6)	474 (25.0)	522 (17.8)	182 (11.0)	41 (9.4)	
Current residence, N (%)						<0.001
Urban	2378 (34.4)	814 (42.9)	1005 (34.2)	450 (27.3)	109 (24.9)	
Rural	4543 (65.6)	1083 (57.1)	1932 (65.8)	1199 (72.7)	329 (75.1)	
Parental education level, N (%)						<0.001
Illiteracy	5434 (78.5)	1413 (74.5)	2317 (78.9)	1334 (80.9)	370 (84.5)	
Primary school or above	1487 (21.5)	484 (25.5)	620 (21.1)	315 (19.1)	68 (15.5)	
Education level, N (%)						<0.001
No formal education	2962 (42.8)	498 (26.3)	1255 (42.7)	921 (55.9)	288 (65.8)	
Elementary school	1577 (22.8)	441 (23.2)	697 (23.7)	350 (21.2)	89 (20.3)	
Middle school	1573 (22.8)	581 (30.6)	666 (22.7)	279 (16.9)	47 (10.7)	
High school and above	809 (11.7)	377 (19.9)	319 (10.9)	99 (6.0)	14 (3.2)	
Employment status N (%)						<0.001
Agricultural employed	3185 (46.0)	722 (38.1)	1354 (46.1)	878 (53.2)	231 (52.7)	
Non-agricultural employed	1694 (24.5)	682 (36.0)	698 (23.8)	270 (16.4)	44 (10.0)	
Retired	1916 (27.7)	471 (24.8)	819 (27.9)	473 (28.7)	153 (34.9)	
Unemployed	126 (1.8)	22 (1.2)	66 (2.2)	28 (1.7)	10 (2.3)	
Drinking status, N (%)						<0.001
Never drink	4071 (58.8)	910 (48.0)	1764 (60.1)	1088 (66.0)	309 (70.5)	
Abstainer	543 (7.8)	131 (6.9)	220 (7.5)	154 (9.3)	38 (8.7)	
Current drinker	2307 (33.3)	856 (45.1)	953 (32.4)	407 (24.7)	91 (20.8)	
Smoking status, N (%)						<0.001
Never smoke	4270 (61.7)	952 (50.2)	1828 (62.2)	1161 (70.4)	329 (75.1)	
Former smoker	555 (8.0)	198 (10.4)	235 (8.0)	94 (5.7)	28 (6.4)	
Current smoker	2096 (30.3)	747 (39.4)	874 (29.8)	394 (23.9)	81 (18.5)	
Digestive diseases, N (%)						<0.001
Absence	5244 (75.8)	1622 (85.5)	2295 (78.1)	1094 (66.3)	233 (53.2)	
Presence	1677 (24.2)	275 (14.5)	642 (21.9)	555 (33.7)	205 (46.8)	
Respiratory diseases, N (%)						<0.001
Absence	6233 (90.1)	1799 (93.8)	2663 (90.7)	1448 (87.8)	343 (78.3)	
Presence	688 (9.9)	118 (6.2)	274 (9.3)	201 (12.2)	95 (21.7)	
Arthritis or rheumatism, N (%)						<0.001
Absence	4585 (66.2)	1551 (81.8)	1992 (67.8)	875 (53.1)	167 (38.1)	
Presence	2336 (33.8)	346 (18.2)	945 (32.2)	774 (46.9)	271 (61.9)	
Cardio-metabolic diseases, N (%)						<0.001
Absence	4400 (63.6)	1328 (70.0)	1858 (63.3)	975 (59.1)	239 (54.6)	
Presence	2521 (36.4)	569 (30.0)	1079 (36.7)	674 (40.9)	199 (45.4)	

Notes: *p* value determined using χ^2^ test or analysis of variance F-test. ACEs, adverse childhood experiences; N, number; SD, standard deviation.

**Table 2 healthcare-12-02539-t002:** Association between ACE scores and depressive symptom trajectories by multinominal logistic regression.

Variables	Continuing-Low-to-Middle vs. Continuing Low		Continuing-Middle-to-High Vs. Continuing Low		Continuing-High vs. Continuing Low	
	Model 1	Model 2	Model 1	Model 2	Model 1	Model 2
	OR (95%CI)	OR (95%CI)	OR (95%CI)	OR (95%CI)	OR (95%CI)	OR (95%CI)
ACE scores						
1	1.252 (1.084, 1.446)	1.237 (1.070, 1.431)	1.482 (1.240, 1.771)	1.435 (1.195, 1.723)	2.015 (1.450, 2.800)	1.895 (1.353, 2.654)
2	1.450 (1.221, 1.723)	1.414 (1.187, 1.683)	2.081 (1.694, 2.557)	1.927 (1.559, 2.382)	4.816 (3.439, 6.745)	4.172 (2.949, 5.902)
3	1.948 (1.524, 2.488)	1.885 (1.470, 2.416)	3.899 (2.966, 5.126)	3.504 (2.642, 4.648)	7.836 (5.191, 11.827)	6.290 (4.108, 9.632)
≥4	2.407 (1.633, 3.550)	2.145 (1.448, 3.178)	7.458 (4.999, 11.127)	5.966 (3.957, 8.993)	20.219 (12.115, 33.744)	14.177 (8.350, 24.071)
Chronic diseases						
Digestive diseases		1.498 (1.272, 1.764)		2.422 (2.019, 2.904)		3.760 (2.921, 4.840)
Respiratory diseases		1.496 (1.181, 1.894)		1.926 (1.476, 2.513)		3.485 (2.487, 4.884)
Arthritis or rheumatism		1.760 (1.520, 2.038)		2.798 (2.371, 3.302)		4.252 (3.326, 5.436)
Cardio-metabolic diseases		1.292 (1.130, 1.478)		1.484 (1.266, 1.741)		1.554 (1.221, 1.978)

Notes: The reference group of the multinomial logistic regression model was the continuing-low group. Model 1 was adjusted for gender, age, education level, marital status, hukou status, residence, parental education level, participants’ education level, participants’ employment status, smoking, and drinking in 2011 baseline survey. Model 2 additionally included the mediators. ACEs, adverse childhood experiences; OR, odds ratio; CI, confidence interval.

**Table 3 healthcare-12-02539-t003:** KHB mediation analysis of different chronic diseases on the association between ACE scores and depressive symptom trajectories.

ACE Scores	Continuing-Low-to-Middle vs. Continuing Low				Continuing-Middle-to-High vs. Continuing Low				Continuing-High vs. Continuing Low			
	1	2	3	≥4	1	2	3	≥4	1	2	3	≥4
	OR (95%CI)	OR (95%CI)	OR (95%CI)	OR (95%CI)	OR (95%CI)	OR (95%CI)	OR (95%CI)	OR (95%CI)	OR (95%CI)	OR (95%CI)	OR (95%CI)	OR (95%CI)
Total effect	1.277 (1.104, 1.477)	1.528 (1.283, 1.819)	2.132 (1.662, 2.734)	2.626 (1.772, 3.891)	1.527 (1.271, 1.834)	2.236 (1.808, 2.765)	4.447 (3.351, 5.901)	8.735 (5.789, 13.179)	2.080 (1.485, 2.914)	5.210 (3.682, 7.371)	9.030 (5.897, 13.827)	25.402 (14.948, 43.165)
Direct effect	1.237 (1.070, 1.431)	1.414 (1.187, 1.683)	1.885 (1.470, 2.416)	2.145 (1.448, 3.178)	1.435 (1.195, 1.723)	1.927 (1.559, 2.382)	3.504 (2.642, 4.648)	5.966 (3.957, 8.993)	1.895 (1.353, 2.654)	4.172 (2.949, 5.902)	6.290 (4.108, 9.632)	14.177 (8.350, 24.071)
Indirect effect	1.032 (0.960, 1.110)	1.081 (1.004, 1.164)	1.131 (1.048, 1.221)	1.224 (1.126, 1.330)	1.064 (0.930, 1.218)	1.160 (1.013, 1.329)	1.269 (1.106, 1.456)	1.464 (1.270, 1.688)	1.098 (0.899, 1.340)	1.249 (1.021, 1.526)	1.435 (1.171, 1.759)	1.792 (1.454, 2.208)
Mediators	Mediation (%)				Mediation (%)				Mediation (%)			
Total indirect influence	12.91	18.30	16.29	20.93	14.67	18.48	15.96	17.59	12.72	13.46	16.43	18.03
Digestive diseases	5.46	5.70	5.92	6.33	6.90	6.56	6.57	6.16	5.97	4.79	6.67	6.19
Arthritis or rheumatism	6.65	10.83	8.20	9.89	6.99	10.38	7.57	8.02	5.68	7.12	7.22	7.56
Respiratory diseases	N.A.	N.A.	2.62	4.52	N.A.	1.84	2.17	3.28	N.A.	1.71	2.80	4.19
Cardio-metabolic diseases	N.A.	N.A.	N.A.	N.A.	N.A.	N.A.	N.A.	N.A.	N.A.	N.A.	N.A.	N.A.

Notes: Model 1 was adjusted for gender, age, education level, marital status, hukou status, residence, parental education level, participants’ education level, participants’ employment status, smoking, and drinking in 2011 baseline survey. Model 2 additionally included the mediators. ACEs, adverse childhood experiences; OR, odds ratio; CI, confidence interval; N.A., not applicable.

## Data Availability

The original data presented in this study are openly available at http://charls.pku.edu.cn/en (accessed on 16 September 2024). The datasets used and analyzed during the current study are available from the corresponding author on reasonable request.

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
