# Peer review of "The Mediating Effects of Chronic Diseases in the Relationship Between Adverse Childhood Experiences and Trajectories of Depressive Symptoms in Later Life: A Nationwide Longitudinal Study"

_healthcare, 2024, doi:10.3390/healthcare12242539_

Round 1

Reviewer 1 Report

Comments and Suggestions for Authors

This is a tremendous study in its scope and importance. The authors don't explain the clear and obvious link between ACES, depression, illness, and digestive problems until the discussion: the stress response. Murri et al. (2014) meta-analysis shows one of most consistent findings with depression is dysregulation of HPA axis. A key component of that is an overly reactive amygdala, see Roberson-Nay et al. (2006) and Yang et al. (2010). There is plenty of animal research to demonstrate the effects of early-life and pre-natal stress on a larger amygdala and smaller hippocampus (key for regulation via a negative feedback loop), see Salm et al. (2004) and Mueller & Bale (2008). And that neurogenesis in the hippocampus is crucial for alleviation of depressive symptoms (Santarelli et al., 2003). Chronically high glucocorticoids levels are also linked to immune system suppression (Glaser & Glaser, 2005; Dhabhar, 2009) as well as the gut microbiome (Clapp et al. 2017).

In fact, in the abstract the authors state "the underlying mechanisms are still not well understood." I would strongly argue that is not the case, but a further exploration of the nature of the interactive relationship between ACES, illness, and depression is important and can inform future treatment.

The longitudinal nature and size of this study contributes greatly to the field as another strong piece of evidence as to the negative impacts that early-life stressors have on the life-history of an individual. When I read this, I see a very clear through-line of ACES to depression/illness/GI problems, and I would just like the authors to make this link more clearly before the discussion. I think this is also important to discuss as it informs the attempt to show a mediational relationship between several of these variables.

Author Response

Comment 1: This is a tremendous study in its scope and importance. The authors don't explain the clear and obvious link between ACES, depression, illness, and digestive problems until the discussion: the stress response. Murri et al. (2014) meta-analysis shows one of most consistent findings with depression is dysregulation of HPA axis. A key component of that is an overly reactive amygdala, see Roberson-Nay et al. (2006) and Yang et al. (2010). There is plenty of animal research to demonstrate the effects of early-life and pre-natal stress on a larger amygdala and smaller hippocampus (key for regulation via a negative feedback loop), see Salm et al. (2004) and Mueller & Bale (2008). And that neurogenesis in the hippocampus is crucial for alleviation of depressive symptoms (Santarelli et al., 2003). Chronically high glucocorticoids levels are also linked to immune system suppression (Glaser & Glaser, 2005; Dhabhar, 2009) as well as the gut microbiome (Clapp et al. 2017).

Response 1: Thank you very much for your recognition of our research and for pointing out this issue. ACEs contribute to stress responses, which in turn can promote the occurrence of depression and illness. We are grateful for the evidence and references you provided, and we have incorporated them to our introduction (page 2, line 84-90; page 2-3, line 94-98;): “Many clinical studies have shown that depression is associated with a highly dysregulated hypothalamic-pituitary-adrenal (HPA) axis and abnormal amygdala function [17-19]. Numerous animal experiments have also demonstrated that early experiences and prenatal stress have a certain impact on the enlargement of the amygdala and the shrinkage of the hippocampus [20, 21]. The stimulation of neurogenesis in the hippo-campus is crucial for alleviating depression [22]. ... The stress response may suppress immune function, and in some cases, enhance it. It may also communicate bidirectionally with the central nervous and endocrine systems through the immune system, causing dysbiosis and inflammation of the gut microbiota, thereby affecting human health [24-26].”.

Comment 2: In fact, in the abstract the authors state “the underlying mechanisms are still not well understood.” I would strongly argue that is not the case, but a further exploration of the nature of the interactive relationship between ACES, illness, and depression is important and can inform future treatment.

Response 2: We really appreciate your valuable comment. As you pointed out, our previous statement in the abstract that “the underlying mechanisms are still not well understood.” was indeed inappropriate. We have revised it into “the interactive relationship between ACEs, depression and chronic diseases are still not well understood.” (page 1, line 18-19)

Comment 3: The longitudinal nature and size of this study contributes greatly to the field as another strong piece of evidence as to the negative impacts that early-life stressors have on the life-history of an individual. When I read this, I see a very clear through-line of ACES to depression/illness/GI problems, and I would just like the authors to make this link more clearly before the discussion. I think this is also important to discuss as it informs the attempt to show a mediational relationship between several of these variables.

Response 3: Thank you for recognizing the significance of our research and for your helpful comments. According to your suggestion, we further emphasized the connections between ACEs, depression, chronic diseases, and digestive diseases, as well as the mediating role of chronic diseases, particularly digestive diseases, in the relationship between ACEs and depression in our discussion (page 12, line 435-441): “A number of studies have demonstrated that ACEs are associated with the development of digestive system abnormalities such as irritable bowel syndrome, gastrointestinal symptoms, and gut-brain interaction disorders [69-71]. These gastrointestinal diseases can further lead to depression through the HPA axis, intestinal permeability, gut microbiota, and inflammation [72]. Additionally, individuals with gastrointestinal dysfunction may experience abnormalities in brain regions, such as significant activation of the amygdala, which increases the risk of depression [73].”.

Reviewer 2 Report

Comments and Suggestions for Authors

Peer review of the paper called “The mediating effect of chronic diseases in the relationship between adverse childhood experiences and trajectories of depressive symptoms in later life: A nationwide longitudinal study.”

First of all, I would like to acknowledge the excellent work the research team has put to write this paper! The findings are crucial for the field of ACE and its consequences on the physical and mental health of individuals.

Abstract:

Great abstract

Introduction:

The authors have done a great job in writing their Introduction. The argument flows very well and is supported by excellent quality references.

Methods:

Very impressive design and strong data set for a longitudinal study.

Can the authors cite the name of the ACE scale they used? They mention “the most widely used ACE scale.”

Line 166: there is a typo: AECs instead of ACEs.

Any references for lines 171 to 175?

Results:

Overall, very strong statistical analysis.

Line 201: Tense: ACEs scores are presented… (this is true now).

Refer in the text to Supplementary 3 for information regarding baseline characteristics and ACEs.

Fix the title of Figure 2.

Line 232: … are reported in Table 1.

Line 273: … This preliminary (missing word) suggested.

Discussion

Line 338: ACEs

One limitation that you should mention is that the chronic disease assessment was based on the 2011 survey and not reassessed through the years. What about people diagnosed with chronic diseases between 2011 and 2018, as you considered this timeframe to measure depression?

I think the discussion would benefit from a whole paragraph on the implications of such findings. The authors mention it briefly (one sentence), but how does it look on a broader scale? What do these findings mean in the field of adverse childhood experiences and their impact on physical and mental health?

I would also recommend rephrasing the limitation surrounding ACEs in lines 452 to 454. First, you did not include a measure of sexual abuse, which is one of the most traumatic ACEs a child can experience. Is parental separation as traumatic as some of the other ACEs presented? I am not too sure, and this has been a limitation of many ACE measures that should be acknowledged.

Author Response

Reviewer 2: Peer review of the paper called “The mediating effect of chronic diseases in the relationship between adverse childhood experiences and trajectories of depressive symptoms in later life: A nationwide longitudinal study.”

First of all, I would like to acknowledge the excellent work the research team has put to write this paper! The findings are crucial for the field of ACE and its consequences on the physical and mental health of individuals.

Abstract:

Great abstract

Introduction:

The authors have done a great job in writing their Introduction. The argument flows very well and is supported by excellent quality references.

Methods:

Very impressive design and strong data set for a longitudinal study.

Comment 1: Can the authors cite the name of the ACE scale they used? They mention “the most widely used ACE scale.”

Response 1: Thank you so much for your appreciation of our research and for your valuable comments. We systematically reviewed the literature on ACEs measurements. In different studies, different ACE scales were used, with the number of these indicators ranging from 1 to 20, or even more (Finkelhor and Shattuck et al., 2013; Wolf and Suntheimer, 2019; Ma and Ji et al., 2024). In summary, there is currently no consensus on a standardized ACE scale. Consequently, we have meticulously referenced previous high-quality research to select 10 widely recognized indicators (Felitti and Anda et al., 1998; Lin and Wang et al., 2021; Lin and Cao et al., 2022; Huang and Li et al., 2024; Shi and Kou et al., 2025). In order to avoid an arbitrary viewpoint, we have modified the sentences in our article (page 4, line 151): “we extracted 10 widely used ACE indicators,”.

References:

Finkelhor, D. and A. Shattuck, et al. (2013). Improving the adverse childhood experiences study scale. JAMA Pediatr 167 (1): 70-75.

Wolf, S. and N. M. Suntheimer (2019). A dimensional risk approach to assessing early adversity in a national sample. Journal of Applied Developmental Psychology 62: 270-281.

Ma, N. and X. Ji, et al. (2024). Adverse childhood experiences and mental health disorder in China: A nationwide study from CHARLS. Journal of Affective Disorders 355: 22-30.

Felitti, V. J. and R. F. Anda, et al. (1998). Relationship of childhood abuse and household dysfunction to many of the leading causes of death in adults. The Adverse Childhood Experiences (ACE) Study. Am J Prev Med 14 (4): 245-258.

Lin, L. and H. H. Wang, et al. (2021). Adverse Childhood Experiences and Subsequent Chronic Diseases Among Middle-aged or Older Adults in China and Associations With Demographic and Socioeconomic Characteristics. JAMA Network Open 4 (10): e2130143.

Lin, L. and B. Cao, et al. (2022). Association of Adverse Childhood Experiences and Social Isolation With Later-Life Cognitive Function Among Adults in China. JAMA Network Open 5 (11): e2241714.

Huang, R. and S. Li, et al. (2024). Adverse childhood experiences and falls in older adults: The mediating role of depression. Journal of Affective Disorders 365: 87-94.

Shi, S. and W. Kou, et al. (2025). The impact of adverse childhood experiences on cognitive function among middle-aged and older Chinese adults: Multiple mediators of cognitive reserve and depressive symptoms. Journal of Affective Disorders 368: 258-265.

Comment 2: Line 166: there is a typo: AECs instead of ACEs.

Response 2: Thank you for your meticulous review. We have corrected the typo (page 5, line 189).

Comment 3: Any references for lines 171 to 175?

Response 3: Thank you for your professional review. We have added relevant references (page 5, line 198-199): “[45] Nagin, D.S., C.L. Odgers, Group-based trajectory modeling in clinical research. Annu Rev Clin Psychol. 2010. 6:109-138. doi: 10.1146/annurev.clinpsy.121208.131413. [46] Nagin D. 2005. Group-based modeling of development. Cambridge, MA: Harvard Univ Press.”.

Comment 4: Line 201: Tense: ACEs scores are presented… (this is true now).

Response 4: Thank you for this valuable comment. We have modified the tense (page 6, line 225).

Comment 5: Refer in the text to Supplementary 3 for information regarding baseline characteristics and ACEs. Fix the title of Figure 2.

Response 5: We sincerely appreciate your suggestion. We have changed the title of Figure 2 to “The percentage of participants with different ACE scores and ACE types” (page 6, line 243).

Comment 6: Line 232: … are reported in Table 1.

Response 6: Thank you for this valuable comment. We have modified the tense (page 7, line 256).

Comment 7: Line 273: … This preliminary (missing word) suggested.

Response 7: Thank you for your meticulous review. We have added the missing word “finding” (page 9, line 296).

Comment 8: Line 338: ACEs

Response 8: Thank you for pointing this out. We have changed “ACE” to “ACEs” (page 11, line 361).

Comment 9: One limitation that you should mention is that the chronic disease assessment was based on the 2011 survey and not reassessed through the years. What about people diagnosed with chronic diseases between 2011 and 2018, as you considered this timeframe to measure depression?

Response 9: Thank you for this helpful comment and we fully agree with your viewpoint. We selected the information of chronic diseases at baseline primarily to ensure that the timing of the mediators precedes the occurrence of the dependent variable (depressive trajectory), thereby contributing to a more reliable causal relationship. Accordingly, we have added this limitation to our discussion (page 13, line 486-491): “Thirdly, the measurement of chronic diseases was based on data from the 2011 base-line survey, and our study did not take into account the population diagnosed with chronic diseases from 2011 to 2018. In fact, new-onset chronic diseases may lead to a deterioration in mental health, thereby affecting the trajectories of depressive symptoms [80]. Further in-depth research is needed to clarify these causal relationships in the future.”.

Comment 10: I think the discussion would benefit from a whole paragraph on the implications of such findings. The authors mention it briefly (one sentence), but how does it look on a broader scale? What do these findings mean in the field of adverse childhood experiences and their impact on physical and mental health?

Response 10: Thank you for this valuable comment. According to your suggestions, we have added another paragraph in the discussion to emphasize the implications of our findings (page 14, line 500-518): “This study disclosed the relationship between ACEs and the depression trajectory among middle-aged and elderly individuals in China, and probed into the mediating role of various chronic diseases. This discovery holds profound implications for formulating public health policies and enhancing clinical practice. Although the causes of depression among middle-aged and elderly people can be partly ascribed to the onset of chronic diseases, their risk factors can frequently be traced back to childhood. Hence, this study emphasized the necessity for all sectors of society to pay attention to the environment in which children grow up and called upon the government to formulate relevant policies to shield children from the harm of ACEs, which is of vital importance for their long-term physical and mental well-being. Given the aging population and the escalating incidence of chronic diseases, it is anticipated that the incidence rate of de-pression among the middle-aged and elderly will continue to rise. Our research pro-vided a scientific basis for identifying potential prevention and intervention targets for depression. Strengthening the management of chronic diseases such as arthritis, rheumatoid arthritis, digestive system diseases, and respiratory system diseases among middle-aged and elderly people, and developing comprehensive medical service measures, is of great significance for alleviating psychological trauma caused by ACEs and reducing the risk of depression in later life.”.

Comment 11: I would also recommend rephrasing the limitation surrounding ACEs in lines 452 to 454. First, you did not include a measure of sexual abuse, which is one of the most traumatic ACEs a child can experience. Is parental separation as traumatic as some of the other ACEs presented? I am not too sure, and this has been a limitation of many ACE measures that should be acknowledged.

Response 11: Thank you for your helpful comment. We agree with your advice and have rephrased the limitation about ACEs (page 13-14, line 491-496): “Finally, the severity of trauma caused by different types of ACEs may vary. However, we did not take into account the weight of each ACE entry, instead assuming that their risks were equal and accumulated ACEs scores were used for statistical analysis, which may affect the accuracy of the measurements [81]. Meanwhile, we did not incorporate an assessment of sexual abuse, which is one of the most traumatic ACEs that children may endure.”.

Reviewer 3 Report

Comments and Suggestions for Authors

Thank you for the opportunity to review this study entitled "The Mediating Effects of Chronic Diseases in the Relationship between Adverse Childhood Experiences and Trajectories of Depressive Symptoms in Later Life: A Nationwide Longitudinal Study” (healthcare-3343387).

The paper presents an exploration of the association between adverse childhood experiences (ACEs) and the development of depression in later life. The mediating roles of chronic diseases in this relationship were also investigated. The sample included 6921 participants aged 45 and older. The China Health and Retirement Longitudinal Study (CHARLS) data from 2011, 2013, 2015, and 2018, combined with the 2014 life history survey, have been used.

In my opinion, the research topic is relevant, and the study is interesting. The large sample size is undoubtedly a great strength of this study. Another extremely relevant aspect is the longitudinal research design. This is an indisputably valuable factor that allows obtaining beneficial results for scientific research. Parallelly, some issues need to be addressed before the paper will be suitable for publication.

  • Abstract: Please add information about the sample (e.g., the percentage of males and females, mean age, and SD) to provide a clearer picture of the study's demographics.
  • Introduction: The first section on depression is the starting point of the study, and it serves as a factor that justifies its importance. Precisely in light of this important role, it should be further expanded. The authors should make a further effort to describe the problem of depression and highlight the importance of deepening its risk factors.
  • Introduction: The research gap should be described more explicitly, to give further relevance to this research.
  • Introduction: The authors stated: “We proposed the following hypothesis that exposure to a higher number of ACEs would [..]”. To encourage greater reading fluency, it should be: “Based on the presented scientific literature, we hypothesized that exposure [..]” or “We proposed the following hypothesis: the exposure to […]”
  • Discussion: The discussions are very detailed and well-structured. Perhaps, given the length, it could be useful to divide them into paragraphs to support the reader in reading specific results of interest.
  • Conclusions: This section should be expanded by further developing the practical implications of this study.

Best wishes

Author Response

Reviewer 3: Thank you for the opportunity to review this study entitled "The Mediating Effects of Chronic Diseases in the Relationship between Adverse Childhood Experiences and Trajectories of Depressive Symptoms in Later Life: A Nationwide Longitudinal Study” (healthcare-3343387).

The paper presents an exploration of the association between adverse childhood experiences (ACEs) and the development of depression in later life. The mediating roles of chronic diseases in this relationship were also investigated. The sample included 6921 participants aged 45 and older. The China Health and Retirement Longitudinal Study (CHARLS) data from 2011, 2013, 2015, and 2018, combined with the 2014 life history survey, have been used.

In my opinion, the research topic is relevant, and the study is interesting. The large sample size is undoubtedly a great strength of this study. Another extremely relevant aspect is the longitudinal research design. This is an indisputably valuable factor that allows obtaining beneficial results for scientific research. Parallelly, some issues need to be addressed before the paper will be suitable for publication.

Comment 1: Abstract: Please add information about the sample (e.g., the percentage of males and females, mean age, and SD) to provide a clearer picture of the study's demographics.

Response 1: We sincerely appreciate your professional review and fully agree with this valuable comment. We have added detailed information about the sample (page 1, line 28-29): “The age of the 6921 participants was 57.2 ± 8.0 years, with females comprising 53.9% and males 46.1%.”

Comment 2: Introduction: The first section on depression is the starting point of the study, and it serves as a factor that justifies its importance. Precisely in light of this important role, it should be further expanded. The authors should make a further effort to describe the problem of depression and highlight the importance of deepening its risk factors.

Response 2: Thank you for this valuable comment. As you pointed out, we have further expanded the description of the current status of depression and emphasized the importance of deepening its risk factors (page 2, line 54-62): “Depression severely limits psychosocial functioning, diminishes the quality of life, and imposes a significant social burden. As a major public health issue, it has led to an escalating burden of disease, particularly within the elderly population [4]. However, the treatment rate for depression is alarmingly low, with many individuals failing to receive adequate therapy [2]. A study reported that in low-and middle-income countries, over 75% of individuals with depression are unable to get treatment [5]. Consequently, there is an urgent need to identify risk factors and potential mechanisms associated with depression to facilitate the timely implementation of preventative and interventional strategies.”.

Comment 3: Introduction: The research gap should be described more explicitly, to give further relevance to this research.

Response 3: Thank you for your helpful comment. We have added a description of the research gap in the introduction to further highlight the significance of our research (page 3, line 108-115): “Previous studies have indicated that ACEs can increase the risk of depression in middle-aged and elderly individuals [11-15], and many studies have shown an association between ACEs and various chronic diseases [27-31]. Currently, there is no research exploring whether different types of chronic diseases mediate the relationship between ACEs and depression trajectories. A study found that multimorbidity plays a mediating role between ACEs and depression in middle-aged and elderly people, but it did not clarify the relationship between ACEs and depression trajectory, and it did not consider the mediating effects of different types of chronic diseases [35].”.

Comment 4: Introduction: The authors stated: “We proposed the following hypothesis that exposure to a higher number of ACEs would [..]”. To encourage greater reading fluency, it should be: “Based on the presented scientific literature, we hypothesized that exposure [..]” or “We proposed the following hypothesis: the exposure to […]”

Response 4: Thank you for your valuable advice. According to your suggestion, we have changed the sentence to “Based on the presented scientific literature, we hypothesized that exposure [...]” (page 3, line 119-120).

Comment 5: Discussion: The discussions are very detailed and well-structured. Perhaps, given the length, it could be useful to divide them into paragraphs to support the reader in reading specific results of interest.

Response 5: Thank you for pointing this out. We agree with this comment. Therefore, we have added a subheading to each section in the discussion (line 375, line 403, line 424, line 455, line 500): “4.1. Mechanisms of ACEs Leading to Depression and Chronic Diseases. 4.2. Chronic Diseases Play Mediating Effects between ACEs and Depression Trajectories. 4.3. Mediating Roles of Different Types of Chronic Diseases. 4.4. Strengths and Limitations. 4.5. The Impactions of ACEs on Physical and Mental Health.”.

Comment 6: Conclusions: This section should be expanded by further developing the practical implications of this study.

Response 6: We sincerely appreciate your helpful comment. We have further added the practical implications of our study in the conclusions, emphasizing the importance of our research in the field of the impact of ACEs on their physical and mental health (page 14, line 526-530): “From a comprehensive life-course perspective, we can better understand and prevent the physical and mental health problems that stem from ACEs. Public health measures aimed at reducing chronic diseases such as arthritis, rheumatism, gastrointestinal dis-eases and pulmonary diseases are an important way to alleviate depression in Chinese middle-aged and elderly people affected by ACEs.”.